# Reducing Bottled Water Use among Adolescents: A Factorial Experimental Approach to Testing the Components of the "Aquatic" Program

**Inga Truskauskaitė-Kunevičienė** [1], **Goda Kaniušonytė** [1,*], **Mykolas Simas Poškus** [1], **Audra Balundė** [1], **Vaida Gabė** [1], **Lina Jovarauskaitė** [1] and **Metin Özdemir** [2]

[1] Institute of Psychology, Mykolas Romeris University, LT-08303 Vilnius, Lithuania; inga.truskauskaite-kuneviciene@fsf.vu.lt (I.T.-K.); mykolas_poskus@mruni.eu (M.S.P.); audra.balunde@mruni.eu (A.B.); gabe.vaida@gmail.com (V.G.); lina.jovarauskaite@fsf.vu.lt (L.J.)

[2] School of Law, Psychology, and Social Work, Örebro University, SE-701 82 Örebro, Sweden; Metin.Ozdemir@oru.se

\* Correspondence: godakan@mruni.eu

**Abstract:** The aim of the current study was to assess the components of the intervention program "Aquatic", targeted at the reduction of bottled water use in adolescence. The Comprehensive Action Determination Model was chosen as a theory of change for the development and evaluation of pro-environmental behavior intervention. We examined the impact of five experimental intervention factors (water bottles, promo video, prompts, goal setting, and feedback) on eight intervention program outcomes: Perceived behavioral control, Social norm, Habit, Awareness of need, Awareness of consequence, Personal norm, Intention, and Behavior. The study sample consisted of 419 adolescents (52.8% girls, $M_{age}$ = 15.21, $SD_{age}$ = 0.64) from Lithuania. A factorial experimental study design was used, and a Latent change modeling approach was applied for the evaluation of individual and combined effects of intervention components. *Promo video*, *Prompts*, and *Goal setting* had a positive effect on Awareness of consequence, Social norm, and Awareness of need, respectively. Receiving a *Water bottle* in combination with *Promo video* had a positive effect on Perceived behavioral control and in combination with *Prompts* as well as *Goals*—on Awareness of need. *Water bottles*, *Promo-video*, *Prompts*, and *Goals*, but not *Feedback*, had value in the promotion of targeted pro-environmental outcomes.

**Keywords:** pro-environmental behavior; factorial experiment; adolescence; intervention

## 1. Introduction

Modern lifestyles and especially ever-increasing consumption leave growing piles of trash that cause dramatic consequences to the planet and, hence, to the well-being of people [1]. Every year, human beings produce around two billion tons of household waste, and this number is gradually increasing [2]. In 2016 only, humanity produced 242 million tons of plastic waste [3]. Despite recycling efforts, every year 150,000 to 500,000 tons of plastic waste from the EU end up in the oceans [4]. It does not only literally threaten the lives of sea animals, but also, through a consumption cycle, comes back to people and may end up in their bodies, potentially causing severe health issues [4]. It becomes increasingly obvious that focusing only on recycling is not the solution and the World should move towards drastic reduction of single use plastic such as water bottles [5]. The current study reports on the possibilities for reduction of bottled water use among adolescents by delivering the intervention program "Aquatic" that was developed based on the theory of change grounded in the Comprehensive Action Determination Model (CADM) [6]. The intervention components were chosen based on the review of pro-environmental intervention strategies by Osbaldiston and Schott [7] and include such strategies as making it easy, social modeling, reminder-prompts, goal setting, and providing environment-related feedback.

## 1.1. Conceptual Models on Promoting Pro-Environmental Behaviors

Although national and international strategies are seen as the primary and most effective way of solving the problem of consumption-related waste [4], individual efforts and, especially, changing consumption routines as well as promoting alternative acceptable behaviors within nature's resource constraints, is among the important contributors to the planet's and human welfare [8]. Less than a decade ago, the Comprehensive Action Determination Model (CADM) [6] was suggested as a theory of behavioral change, which is claimed to be especially beneficial for pro-environmental behavior interventions. The CADM incorporates such widely known behavior-related theoretical models as the Norm-Activation Model (NAM) [9] and the Theory of Planned Behavior (TPB) [10] and encompasses normative (awareness of consequences, awareness of need, social norm, and personal norm), intentional, situational (perceived behavioral control and access to behavior), and habitual determinants of individual behavior. In the CADM, perceived consequences of behavior (awareness of consequences), perceived need to behave in a pro-environmental manner (awareness of need), as well as perceived social pressure (social norms) lead to a moral obligation to act pro-environmentally (personal norm). Personal norms, in turn, affect intention and habit to perform the behavior. Furthermore, intentions may be determined not only by personal norms but also by the social norms and perceived behavioral control. In CADM, both perceived behavioral control and access to behavior influence the habit and behavior directly. Moreover, a habit might act as a direct predictor of behavior.

## 1.2. Pro-Environmental Behaviors among Adolescents

Although the CADM model has been extensively used in the literature for explaining adult pro-environmental behavior, for example [11,12], there is a lack of studies that address adolescents' pro-environmental behavior in a CADM framework. Although youth have less influence on household consumption decisions than adults, they still can affect it through their parents [13]. Additionally, by now it has been well acknowledged that adolescents already have considerable spending power, are actively engaged in consumption behavior, and make consumption-related decisions on a daily basis [14]. Moreover, youth is the next generation of adults with consumer power [15] as well as being future decision-makers that will need to address the nature-related challenges and deal with the consequences of the behavior of earlier generations. More than a decade ago, the United Nations clearly stated that school children and adolescents are an important age group for applying pro-environmentally oriented behavioral change [16]. Youth continues to be an important target group for promoting pro-environmental behaviors, especially having in mind that they are at a life stage where they are forming their system of beliefs, morals, and values, including the prosocial ones [17].

Over the last decades, there has been a substantial effort to promote individual pro-environmental behaviors and there is a solid body of literature related to changing adult environmentally friendly behaviors, for a review see [7]. However, the literature on fostering adolescent pro-environmental efforts is rather scarce and focused mainly on knowledge-based educational training, for example [18,19]. Nevertheless, there is some evidence from research on adolescents that when seeking to influence youth's pro-environmental behavior it would be meaningful to address social norms and perceived behavioral control [20]. Also, when fostering adolescents' behavioral intention, awareness of consequence alongside the personal norm and perceived behavioral control may play an important role [21]. Additionally, in one of the few experimental studies on youth, it has been found that goal-setting-related techniques are the vehicle for behavioral change in adolescence, namely for promoting energy-saving behavior [22].

## 1.3. The Current Study

The goal of the current study is to test the effectiveness of the components of an intervention program "Aquatic" that aims to promote pro-environmental behavior, namely,



the reduction of bottled water use among adolescents. The development of the intervention was informed by the Multiple Phase Optimization Strategy (MOST) approach [23], as it allowed the comprehensive evaluation of multiple components in the behavioral intervention. We used a factorial experiment for the evaluation of individual and combined intervention components, as it is a comprehensive method that is most appropriate when conducting research to select components for inclusion in a multicomponent intervention, and it allows doing it with sufficient statistical power with a reasonable sample size [24,25].

Perceived behavioral control and social norms were chosen as the main targets of the intervention, as, based on CADM [6], the two constructs represent the agents (or mediators) of change and are expected to accumulate the processual interplay between the factors important for actualizing the particular behavior and, eventually, should lead to it. To promote these outcomes, we chose five intervention strategies, namely, making it easy, social modeling, prompts, goal setting, and providing feedback [7].

To increase perceived behavioral control, we used one of the proven strategies of pro-environmental behavioral change, namely, *Making it easy*, as the first intervention component. Therefore, we provided the participants with reusable water bottles. As it was found that the current strategy works best when paired with *Prompts* [7], we added prompts as the second intervention condition. Everyday app-based reminders to pour some tap water were used as prompts. We also expected that prompts will assist in breaking the habit of bottled water use, which, as found by Jovarauskaitė and colleagues [26], is an important predictor of behavior in adolescence. In previous research, when testing the habit discontinuity theory [27], prompts were proven to be an effective strategy for the change of habitual behavior [28]. In our study, receiving a reusable water bottle is seen as a contextual change and, therefore, we expected that prompts would be useful to put them into practice.

To additionally promote perceived behavioral control, we added *Feedback* and *Goal setting* [7] as the third and fourth intervention components. Feedback was found to be effective when given frequently [29], therefore, every time the participants provided information on how much non-bottled water they drank, they had an opportunity to see how it is related to saving the environment and money. As saving the environment was linked to the use of non-bottled water, we expected that feedback would also affect the awareness of consequence. The self-interest-related feedback was included as a motivator for continuity of relatively low-cost and low-effort behavior [30]. We provided person-level and group-level feedback to accumulate possible intervention effects [31]. Goal-setting is generally an understudied but promising strategy for behavioral change [7,22]. Therefore, we also asked the participants to indicate how many bottles of water they purchase weekly and to specify what would be the intended goal for the reduction of the bottled water use. The goal was set personally and not imposed, thus allowing individuals to act naturally and for their innate biases to act in congruence with desirable societal outcomes [32,33].

To promote social norms, we chose *Social modeling* as the fifth intervention component. Social modeling was found to be among the effective strategies for addressing pro-environmental behavior in general [7] and social norms in particular [31]. For modeling the socially acceptable behavior, we used a *Promo video*, where the reusable canteen was suggested as an alternative for purchasing bottled water. As the *Promo video* included some visual elements related to awareness of need and awareness of consequence, we expected additional effects on these constructs.

To our knowledge, no prior experimental study tested the utility of the Comprehensive Action Determination Model in promoting pro-environmental behavior among adolescents. In addition, this study is the first attempt to directly test the effectiveness of each intervention component on the theoretically based mediators of behavioral change. Based on literature analysis, we hypothesized that *Making it easy* alone and combined with *Prompts* would strengthen perceived behavioral control; *Prompts* would promote habit; *Feedback* would foster perceived behavioral control and awareness of consequence; *Goal setting* would promote perceived behavioral control; the *Promo video* would increase



social norms, awareness of need, and consequences. However, due to the novelty of the approach and lack of prior empirical evidence, we tested the effects of each of the five intervention components and their two-by-two combinations on all of the eight related CADM constructs (i.e., perceived behavioral control, social norm, habit, awareness of need, awareness of consequence, personal norm, intention, and behavior).

## 2. Materials and Methods

### 2.1. Design

We used the Multi-phase Optimization Strategy (MOST) screening carried out as $2^5$ factorial experiment for the evaluation of individual and combined effects of intervention components [24,25]. We examined the impact of five experimental intervention factors (each evaluated on two levels) on eight possible intervention program outcomes: Perceived behavioral control, Social norm, Habit, Awareness of need, Awareness of consequence, Personal norm, Intention, and Behavior. The experimental factors of interest were *Making it easy* (reusable water bottles provided by the experimenter vs. no bottles), short *Promo video* about the harm of single-use plastic bottles and benefits of reusable water bottles (watched video vs. no video), *Goal setting* to buy less single-use water bottles in an app (yes vs. no), *Prompts* in an app (reminder notification vs. no notification), and *Feedback* (provided in an app vs. no feedback).

### 2.2. Experimental Factors and Planned Comparisons

In this $2^5$ factorial experiment, five experimental factors were evaluated, each on two levels. The result was a 32-arm factorial experimental design, where, as suggested by Collins and colleagues [24,25], the intervention group is defined by the exposure to the particular component (or combination of components), when the control group, despite the exposure to some conditioning, is not exposed to that particular component (see Table 1). Therefore, the intervention and the control groups differ for the evaluation of each component. We compared the independent effects of each of the five experimental factor levels. In addition, we evaluated the combined effects of all component pairs on the outcomes to assess the effects of both individual and combined exposure to the intervention components. The factors and the actual implementation of the exposure to the intervention components are described below.

#### 2.2.1. Reusable Water Bottle (Making It Easy) Component

The reusable water bottle was given to half of the study sample. We expected that adolescents would carry the water bottle with them and use tap (or well, or spring) water for everyday use. For that purpose, the water bottle had to meet some criteria: (a) it should be functional (lightly weighted, optimal size, etc.), and (b) acceptable (likable) for the adolescents. Therefore, to consult same-age adolescents, we organized two focus groups from different schools that were not the same as the ones chosen for the intervention implementation. The adolescents evaluated 12 different reusable water bottle designs. After consulting the adolescents, clear white PET bottles were chosen. Additionally, participants were asked how they feel about drinking the tap water in their environment and 97.5% responded that they feel neutral or good about it.

#### 2.2.2. Promo Video (Social Modelling) Component

A short promotional video created by the team of the United Nations Environment Program (unep.org) was used. The video was targeted to increasing the awareness of the harm of single-use plastic and modeling possible alternatives to it (the benefits of reusable canteens). The particular video was chosen as it has a humorous component as well as the message is brought through the imitation of romantic relationships, which makes it more attractive to adolescents. Prior to the intervention, during the focus groups, we showed the intended Promo video to the adolescents and asked them to share their emotional reactions to it as well as their thoughts that came to mind after watching it.

**Table 1.** 32-Arm randomization scheme.

| Arm | Water Bottle | Promo Video | Prompts | Feedback | Goal Setting | *n* |
|---|---|---|---|---|---|---|
| 1 | Yes | Yes | Yes | Yes | Yes | 15 |
| 2 | Yes | Yes | Yes | Yes | No | 16 |
| 3 | Yes | Yes | Yes | No | Yes | 25 |
| 4 | Yes | Yes | Yes | No | No | 8 |
| 5 | Yes | Yes | No | Yes | Yes | 10 |
| 6 | Yes | Yes | No | Yes | No | 11 |
| 7 | Yes | Yes | No | No | Yes | 12 |
| 8 | Yes | Yes | No | No | No | 13 |
| 9 | Yes | No | Yes | Yes | Yes | 12 |
| 10 | Yes | No | Yes | Yes | No | 11 |
| 11 | Yes | No | Yes | No | Yes | 15 |
| 12 | Yes | No | Yes | No | No | 7 |
| 13 | Yes | No | No | Yes | Yes | 13 |
| 14 | Yes | No | No | Yes | No | 12 |
| 15 | Yes | No | No | No | Yes | 13 |
| 16 | Yes | No | No | No | No | 13 |
| 17 | No | Yes | Yes | Yes | Yes | 20 |
| 18 | No | Yes | Yes | Yes | No | 25 |
| 19 | No | Yes | Yes | No | Yes | 21 |
| 20 | No | Yes | Yes | No | No | 20 |
| 21 | No | Yes | No | Yes | Yes | 12 |
| 22 | No | Yes | No | Yes | No | 14 |
| 23 | No | Yes | No | No | Yes | 15 |
| 24 | No | Yes | No | No | No | 13 |
| 25 | No | No | Yes | Yes | Yes | 11 |
| 26 | No | No | Yes | Yes | No | 13 |
| 27 | No | No | Yes | No | Yes | 8 |
| 28 | No | No | Yes | No | No | 14 |
| 29 | No | No | No | Yes | Yes | 12 |
| 30 | No | No | No | Yes | No | 12 |
| 31 | No | No | No | No | Yes | 9 |
| 32 | No | No | No | No | No | 11 |

Note. *n* = number of participants in each arm. For each component, "Yes" represents the intervention group and "No" represents the control group.

We cleaned the video of excess information about other sorts of single-use plastic, leaving only parts about single-use water bottles, resulting in the video duration of 1 min., 21 s. A Lithuanian translation was added in the form of subtitles to ensure that all participants would understand the content (video is available on request). To minimize possible distractions and ensure individual attention, the video was shown on tablet computers with earphones. To additionally assure that the video was not shared between participants with different experimental conditions, the Promo video was not available online, and the tablets had no internet access during the exposure.

For delivering the *Prompts*, *Goal setting*, and *Feedback* components (described in the following three subsections), we developed a mobile application for both iOS and Android. We chose a simple water-like design for the mobile application. Different versions of the mobile app were created and distributed to the participants, based on the randomized conditions they were assigned to. Each version of the app had a password for registration. All app users could designate in the app how much non-bottled water they drink and follow the statistics of consumed water. Therefore, even the participants who were not exposed to any of the *Prompts*, *Goal setting*, or *Feedback* components, installed the app onto their mobile phones.

### 2.2.3. Prompts Component

The prompt message contained the short word "pour" together with a water drop emoji. Four *Prompts* messages were tested, and the frequency of the delivery was dis-

cussed during the two focus groups with adolescents. Therefore, during the intervention, adolescents could choose the timing of the prompts by themselves. They could setup up to five different times during the day when they preferred to receive the reminder. The default unremovable setting included one reminder per day. The reminder appeared on the participant's phone at the chosen times as a notification.

### 2.2.4. Feedback Component

The *Feedback* contained three information elements. The kilograms of $CO_2$ saved, the kilograms of plastic waste not produced, and money saved by not buying bottled water. This information was provided in four levels: (1) what is the personal progress of all indicators from the beginning to use mobile application (based on how much of non-bottled water consumed is registered up to date by the user), (2) what would be the personal progress of all indicators after a year if a user would keep drinking tap (or alternatively well or spring instead of purchasing bottled water) water at the same pace (Level 1/days app used $*$ 365 days), (3) what would be the progress of all indicators from the beginning of use of the mobile application if the whole school would keep drinking tap water at the same pace as the user (Level 1 $*$ number of school participants), (4) what would be the progress of all indicators after a year if the whole school would keep drinking tap water at the same pace as the user (Level 3/days app used $*$ 365 days).

### 2.2.5. Goal Setting Component

The *Goal* is to be set immediately after the installation of the app as the first step of the setup. The participants entered the amount of single-use plastic bottles they are purchasing on average during the week. Afterward, they entered the lower number of water bottles they would like to reach for their weekly use. The difference between current and intended use reflected a goal to what amount the participants would like to reduce their weekly purchase of plastic bottles after the program. Adolescents were free to enter any lower number, therefore the goal for the reduction varied across the sample. The participants could see their set up goal in the app all the time. In addition, they could register the amount of single-use water bottles purchased and see how successful they are in keeping up with their goal.

### 2.2.6. Recruitment, Screening, Randomization, and Enrollment

The study protocols were approved by the ethical committee of the Department of Psychology of Mykolas Romeris University, decision No. 2/-2020. The experiment was implemented in a school setting. Schools in Lithuania were selected based on the size of the school and the size of the city they are located in to maximize the resemblance between the intervention participants. Six schools meeting the selection criteria were contacted prior to the intervention. Participants of the intervention were recruited from two high schools that agreed to participate in the experiment. All ninth and tenth graders were eligible to participate in the experiment. Intervention for the optimization purposes and the evaluation (pre-test and post-test) occurred from September 2018 through November 2018. The written parental consents and consents from the adolescents themselves were obtained prior to the first pre-test.

The randomization of the factors was conducted in three levels: school, classroom, and individual, due to different levels of factors. First, due to possible spillover effects and contamination by social modeling, the reusable water bottles were distributed at the school level. Meaning that all participants from one school received water bottles, and participants from the other school—did not. Out of the two selected schools, the school which received water bottles was selected randomly by flipping a coin. Second, promo video and prompts conditions were randomized at the class level. There were 19 classes in total, 10 classes in one school (without water bottles) and 9 in the other (with water bottles). The class in which to show the video was selected randomly by flipping a coin. For prompts, classes were randomized by flipping a coin separately for the promo video groups (those who

watched and those who did not watch the video), resulting in half of the classes getting prompts and half of them not. For the feedback condition, participants were randomized on the individual level within each of the four groups in both schools resulting from the randomization of the previous two conditions (i.e., video/prompts: Yes/Yes, Yes/No, No/Yes, and No/No). For each of the groups, an alphabetical participant list was created and randomized using a random number sequence generated by a computer algorithm. Each sequence then was split in half, one half was exposed to the feedback option in their version of the app and the other one was not. The half that received feedback was decided by flipping a coin. The resulting eight groups (based on video/prompts/feedback exposure/non-exposure) were again split in half (every group separately) for the goal-setting condition, applying the same randomization procedure as for the feedback condition. To balance the age in each condition, the randomization of classes and participants was performed in each cohort (9 and 10 grades) separately, applying the same rules as described above. There were nine classes in ninth grade and ten in tenth grade. To balance the number of participants in each condition, the quota of classes in the promo video condition was set to reflect approximately half of the classes in cohorts; when the coin reached the quota of classes in one of the two conditions, all remaining classes were automatically assigned to the other condition. The total numbers of participants in each condition are presented in Table 2.

**Table 2.** Participants in each condition.

|  | **Yes** | **No** |
|---|---|---|
| Reusable Water bottle |  |  |
| 9th grade | 94 | 131 |
| 10th grade | 112 | 99 |
| Total | 206 | 230 |
| Promo video |  |  |
| 9th grade | 118 | 107 |
| 10th grade | 132 | 79 |
| Total | 250 | 186 |
| Prompts |  |  |
| 9th grade | 119 | 106 |
| 10th grade | 122 | 89 |
| Total | 241 | 195 |
| Feedback |  |  |
| 9th grade | 110 | 115 |
| 10th grade | 109 | 102 |
| Total | 219 | 217 |
| Goal setting |  |  |
| 9th grade | 115 | 110 |
| 10th grade | 108 | 103 |
| Total | 223 | 213 |

*2.3. Intervention*

The intervention program "Aquatic" was developed by the team of the research project "Understanding the psychological mechanisms of the development of pro-environmental behavior in the context of longitudinal intervention (GOGREEN)". Consistent with the CADM, the program was designed to promote the bottled water use-related pro-environmental means by targeting mainly Perceived behavioral control, Habit, Social norm, and Awareness. The intervention implementation consisted of three steps. The first step was the viewing of the short *Promo video* (assigned classrooms only); second, the installation and the setup of the mobile application, which contains different combinations of functionality regarding *Prompts*, *Feedback*, and *Goal setting* (all participants); third, the distribution of reusable water bottles (assigned school only). For the installation of the app, each participant was provided with a registration code for the assigned version of the mobile application. All versions of the mobile app had functionality where users could

register how much water they drank every day from the tap, well, or spring, meaning, not bought in a single-use bottle, and track these statistics. In addition to that, each participant had the functionality they were assigned at the randomization stage (e.g., if they fell in the group with "No" on all conditions: they had only the basic functionality of water consumption registration if they fell in the group with "Yes" on all conditions, they had the basic functionality of water consumption registration and all three conditions: Prompts, Feedback, and Goal setting).

### 2.4. Procedure

During the September—December 2018 period, there were six meetings with participants in schools. First, a presentation of the forthcoming experiment was organized for students in every participating class and for teachers separately. We explained that the purpose of the experiment was to find out what means are effective when promoting environmentally friendly behavior among adolescents. We also distributed printed forms of parental consent and introduced the schedule of our following visits.

The measurements (pre-test and post-test) were conducted during school hours in the classrooms using tablet computers. Depending on the agreement with the schools, either the researchers came to designated classrooms and distributed tablets for filling in surveys, or participants came to the classrooms where researchers were waiting for them with prepared tablets. The pre-test was organized two weeks after the presentation of the experiment. Five days after the pre-test measurement, the intervention was implemented by presenting all components to the participants in the classrooms. The Promo video was shown first (for assigned classes only), then participants were asked to download the app and register with a code assigned to each participant earlier within the randomization stage. It was explained to the participants that all of them have individual entry codes, so the functions of the app may differ, and they were asked not to share information and experience with each other. Finally, reusable water bottles were distributed to the participants from the assigned school as a component of the experiment. To the other school, water bottles were given as a gift after the final measurement. Post-test measurement was carried out one month after the beginning of intervention implementation. Results of the experiment were presented to participants and the school community three weeks after the final measurement.

### 2.5. Participants

This study is part of the longitudinal research project GOGREEN conducted in Lithuania. All the ninth and tenth graders of two schools participated in the factorial experiment. In total, 478 students from both schools were eligible for the study (136 ninth graders and 98 tenth graders from one school, 129 ninth graders and 115 tenth graders from the other school). All students participated in the presentation of the experiment and together with the information about the experiment they received a paper form of parental consent. Students who were not at school on the day of the presentation still had a chance to participate in the experiment as teachers were asked to provide them all the relevant information and the paper form of parental consent as well. Parents of 35 children refused participation in the study, 7 students refused to participate themselves; the resulting 436 participants were included in the randomization stage. If participants were not at school when the conditions of the experiment were administrated, they still had a chance to participate as teachers who had constant support from the researchers were asked to provide participants with relevant personalized information (if assigned—reusable water bottle and assigned app code along with instructions). Four participants had no suitable mobile device; thus, they were excluded from the study. If participants were not at school on the days of the measurements, they were given the link of the questionnaire and could fill it in at home or another day at school.

The attrition (2%) was mainly due to the non-participations in the measurements. However, data was missing completely at random ($\chi^2(27) = 18.37$, $p = 0.89$). Only the

participants with available data on at least one measurement point were included in the analysis. Therefore, the final study sample consisted of 419 participants (52.8% girls, $M_{age}$ = 15.21, $SD_{age}$ = 0.64). The numbers of the participants in each condition for every intervention component are provided in the Section 3.

*2.6. Measures*

As the study assessed components of the behavioral intervention designed to promote pro-environmental behavior (reduction of bottled water use), all outcome variables were specifically related to this particular behavior. To measure the CADM constructs— Perceived behavioral control, Social norm, Habit, Awareness of need, Awareness of consequence, Personal norm, and Intention—we used single-item questions based on the wording of Klöckner and Blöbaum [6]. Using single-item indicators for the measurement of behavioral change-related constructs is a common practice in the field, for example, in [34]. Items were formulated specifically around bottled water purchasing behavior, namely, "People who are important to me expect that I would not buy bottled water" (*social norm*), "Buying bottled water causes many environmental problems" (*awareness of need*), "If I reduce buying bottled water, I contribute to environmental protection" (*awareness of consequences*), "I feel morally obliged not to buy bottled water" (*personal norm*), "I am used to buying bottled water" (*habit*), "It is completely up to me whether I will buy bottled water in the next four weeks" (*perceived behavior control*), "I intend not to use bottled water in the next four weeks" (*intention*). All mentioned items were rated on a five-point Likert scale from 1 (*completely disagree*) to 5 (*completely agree*). Bottled water purchasing behavior was also measured with a single item ("I bought bottled water in the last four weeks"), which was rated from 1 (*never or almost never*) to 5 (*constantly*). In the initial measurement, for Habit and Behavior, a lower score stood for less frequent behavior. However, in the data analysis, the items were recoded so that in all items higher scores represented more desirable outcomes.

*2.7. Data Analysis*

To examine the changes in program outcomes, we used the latent change modelling approach. Latent-variable models provide the possibility to model the measurement error and provide more robust estimates of change over time [35]. In latent change models with two measurement points, the intercept represents the mean level of the measure, and the slope represents the change from baseline to post-test. To have the model identified, we fixed the residuals to zero. To compare the effects of each program component, we dummy-coded the exposure to the intervention component to 1 and non-exposure to 0. We did the same for the combinations of conditions: if the participant was exposed to both of the components, we coded it 1 and if to none of the two components—0; participants who were exposed to either one of the two components were excluded from the analysis. We then regressed the exposure to the intervention component variable on the intercept and the slope of the outcome variables. The regression path from dummy-coded variables to the intercept enabled testing the baseline differences among the intervention and the control conditions. The regression paths from the dummy-coded variables to the slope provided the indication of whether the change in the intervention condition differs from the one in the control condition. Additionally, to contrast the change in the intervention and the control conditions, we ran multi-group latent change models. As the participants of the study were nested into classrooms, to ensure the correctness of the effect sizes calculation [36], the analysis was performed by applying the complex data approach. Based on this approach, the standard errors are computed taking into account complex sampling features [37], namely, participants' clustering in different classrooms. To compare the magnitude of the effects, we computed the Cohen's d effect sizes for the mean change differences in the intervention group and the control group. Based on the guidelines of Gignac and Szodorai [38], the effect size of $\geq$0.1 and <0.2 was considered as small, $\geq$0.2 and <0.3 as moderate, and $\geq$0.3 as large. The effects of intervention components were

considered significant if the regression coefficients were significant at $p < 0.05$ and/or the 95% confidence intervals of the effect sizes did not include 0. All latent difference and multiple group analyses were conducted in Mplus 7.4 [39]. No data imputation was applied. Full information maximum likelihood (FIML) estimator was used in all analyses as a method for taking into account the missing data [40].

## 3. Results

### 3.1. Individual Program Components' Effects on the Intervention Outcomes

We tested the effect of five intervention components, namely, Making it easy (providing a reusable water bottle), Promo video, Prompts, Goal setting, and Feedback on eight outcome measures, that is, Perceived behavioral control, Social norm, Habit, Awareness of need, Awareness of consequences, Personal norm, Intention, and Behavior. All variables across two measurement points were approximately normally distributed, as the coefficients of skewness and kurtosis were within the range of $\pm 2$ [41]. The individual effectiveness results are presented in Table 3. The mean intercepts and slopes for each condition in the intervention and the control groups as well as the baseline comparison between groups can be found in the Supplementary Material Table S1. Further, we will present all study results and will indicate whether our hypotheses were supported, where applicable.

Our hypothesis that receiving a water bottle would strengthen Perceived behavior control was not supported as the effects of receiving a water bottle on Perceived behavior control was small and not significant; the results indicated that receiving water bottles fostered no improvements in any of the program outcomes. Additionally, receiving the water bottle had a moderate reverse effect on Social norms. Specifically, youth who were not provided with water bottles reported an increase in Social norms, when this was not the case for those who did receive water bottles. The baseline level of the Social norm was higher in the intervention group, compared to the control group.

Watching the Promo video promoted a significantly bigger increase in Awareness of consequence, compared to the non-exposure group, which is in line with our hypothesis, however, this effect was found to be small. Also, this was the only significant effect of the Promo video, thus, our hypothesis regarding the potential of the Promo video to foster Social norms and Awareness of need was not confirmed. No baseline differences in Awareness of consequence were found among groups.

The Prompts to fill the water bottle had a moderate effect on Social norms; in particular, the Social norm increased in the intervention group, while the control group demonstrated no change. Additionally, Prompts had a small reverse effect on self-reported Behavior. Specifically, the decrease of bottled water use was bigger in the group that was not exposed to Prompts, compared to the one that was. Intervention and Control groups did not differ in terms of Social norm and Behavior at the baseline. No other significant effects of Prompts were found. Our hypothesis that Prompts should strengthen Habit was not supported, as the effects of Prompts on Habit was small, but not significant.

Setting up the Goal regarding the reduction of buying bottled water had a moderate effect on Awareness of need; those exposed to goal setting reported a bigger increase in Awareness of need compared to the group that did not set up the goal. The control group reported higher initial levels of Awareness of need compared to the intervention group. No other significant effects of Goal setting were revealed. Therefore, the hypothesis that Goal setting should promote Perceived behavioral control was not supported.

**Table 3.** Individual program components' effects on the intervention outcomes.

| | Social Norm | | Awareness on Need | | Awareness of Consequence | | Personal Norm | | Habit | | Perceived Behavioral Control | | Intention | | Behavior | |
|---|---|---|---|---|---|---|---|---|---|---|---|---|---|---|---|---|
| | β | d [95% CI] | β | d [95% CI] | β | d [95% CI] | β | d [95% CI] | β | d [95% CI] | β | d [95% CI] | β | d [95% CI] | β | d [95% CI] |
| Water bottle $n_i$ = 198 $n_c$ = 221 | −0.12 * | −0.24 [−0.43, −0.05] | 0.10 | 0.19 [−0.01, 0.38] | −0.13 | −0.10 [−0.29, 0.09] | −0.19 | −0.15 [−0.34, 0.04] | −0.04 | 0.08 [−0.11, 0.28] | 0.09 | 0.17 [−0.02, 0.36] | −0.01 | −0.02 [−0.22, 0.17] | −0.002 | 0.00 [−0.20, 0.19] |
| Promo video $n_i$ = 237 $n_c$ = 182 | −0.02 | 0.04 [−0.23, 0.16] | 0.02 | 0.03 [−0.23, 0.16] | 0.09 * | 0.19 [−0.01, 0.38] | 0.02 | 0.04 [−0.16, 0.23] | −0.06 | 0.12 [−0.08, 0.31] | 0.07 | 0.12 [−0.08, 0.31] | −0.01 | 0.15 [−0.05, 0.34] | −0.05 | −0.09 [−0.29, 0.10] |
| Prompts $n_i$ = 229 $n_c$ = 190 | 0.12 * | 0.24 [0.05, 0.44] | 0.09 | 0.17 [−0.02, 0.36] | 0.08 | 0.17 [−0.02, 0.36] | 0.01 | 0.02 [−0.17, 0.21] | 0.09 | 0.17 [−0.02, 0.36] | 0.04 | 0.08 [−0.12, 0.27] | −0.05 | −0.10 [−0.29, 0.10] | −0.09 * | −0.19 [−0.38, 0.00] |
| Goal setting $n_i$ = 216 $n_c$ = 203 | 0.04 | 0.07 [−0.12, 0.26] | 0.11 ** | 0.22 [0.03, 0.41] | 0.05 | 0.10 [−0.09, 0.29] | 0.08 | 0.15 [−0.04, 0.35] | −0.05 | 0.10 [−0.09, 0.29] | 0.02 | 0.05 [−0.14, 0.24] | 0.01 | 0.03 [−0.17, 0.22] | 0.07 | 0.13 [−0.06, 0.32] |
| Feedback $n_i$ = 210 $n_c$ = 209 | −0.03 | −0.06 [−0.26, 0.13] | −0.10 | −0.19 [−0.39, 0.00] | −0.11 * | −0.22 [−0.41, −0.03] | 0.004 | 0.01 [−0.18, 0.20] | 0.02 | −0.04 [−0.24, 0.15] | 0.01 | 0.02 [−0.17, 0.22] | −0.12 * | −0.24 [−0.43, −0.04] | −0.14 ** | −0.28 [−0.47, −0.09] |

Note. * $p < 0.05$, ** $p < 0.01$. $n_i$ = number of participants in the intervention group; $n_c$ = number of participants in the control group. Standardized regression coefficients are provided.

Finally, we found that exposure to the Feedback on how much plastic, $CO_2$, and money one saves, had moderate reverse effects on Awareness of consequence, Intention, and Behavior. Specifically, for all three program outcomes, the expected positive change was bigger in the group of youth that did not receive feedback, compared to the one that did. No positive effects of feedback as well as no baseline differences were found. These findings indicate that our hypotheses regarding the positive effects of Feedback on Perceived behavior control and Awareness of consequences were not supported.

### 3.2. Combined Effects of Program Components on the Intervention Outcomes

We also tested the combined effects of the intervention components on all of the program outcomes to investigate whether the combination of components is meaningful. In this approach, the participants who were exposed to the combination of the components are treated as an intervention group and the ones who were not exposed to either of the components as a control group. Therefore, the participants who were exposed to some components but not to others were excluded from the analysis. Due to the decreased sample size and, therefore, the decreased statistical power, we only tested the two-by-two component effects. The combined effectiveness results are presented in Table 4. The intercepts and slopes for each condition in the intervention and the control groups as well as the baseline comparison between groups for the combined effects can be found in the Supplementary Material Table S2.

Receiving a water bottle and watching a Promo video had a large effect on Perceived behavioral control, specifically, an increase in Perceived behavioral control was observed in the intervention group, but not in the control group. From the moderate effect sizes (above 0.2) and also significant within-group effects in the intervention group, compared to non-significant effects in the control group, we can also see that combining a water bottle and a Promo video has the potential for breaking the Habit of purchasing bottled water. The initial level of habit to drink bottled water was higher in the control group, compared to the intervention group. Receiving a water bottle and Prompts to fill the water bottle as well as receiving a water bottle and setting up a Goal had a large effect on Awareness of need; a bigger increase in Awareness of need was observed in the intervention group, compared to the control groups. From the moderate effect sizes, we can also see that combining a water bottle and Prompts has potential for the promotion of Perceived behavioral control, therefore our hypothesis regarding the combined effect of Making it easy and Prompts could not be fully rejected. Moreover, we found that when exposed to Prompts in addition to receiving water bottles, the reverse effect of receiving water bottles on Social norm becomes non-significant. From the analysis of the combined water bottle and Feedback effects, we can see that the reverse effects on Social norm, Awareness of consequence, and Behavior are sustained.

We also discovered that the exposure to Prompts to fill the water bottle in addition to the Promo video reinforced the effect on Awareness of consequence, as the combination of the two components resulted in higher effect size, compared to the individual Promo video effect. However, when combining the two components, the reverse effect on Behavior sustains. No significant effects were found when combining the Promo video and Goal as well as Promo video and Feedback. However, from the moderate effect sizes (above 0.2) we may expect that the combination of Promo video and Goal setting has the potential to promote Awareness of need and consequences as well as break the bottled water use Habit. Also, when combining Promo video and Feedback, the negative effects on Awareness of consequence and Intention become non-significant, when on Behavior, they were sustained.

**Table 4.** Combined (two by two) program components' effects on the intervention program outcomes.

| | Social Norm | | Awareness on Need | | Awareness of Consequence | | Personal Norm | | Habit | | Perceived Behavioral Control | | Intention | | Behavior | |
|---|---|---|---|---|---|---|---|---|---|---|---|---|---|---|---|---|
| | $\beta$ | *d* [95% CI] | $\beta$ | *d* [95% CI] | $\beta$ | *d* [95% CI] | $\beta$ | *d* [95% CI] | $\beta$ | *d* [95% CI] | $\beta$ | *d* [95% CI] | $\beta$ | *d* [95% CI] | $\beta$ | *d* [95% CI] |
| Water bottle + Promo video $n_i$ = 105 $n_c$ = 89 | −0.14 | −0.28 [−0.57, 0.00] | 0.11 | 0.20 [−0.08, 0.48] | 0.04 | 0.20 [−0.20, 0.37] | −0.06 | −0.13 [−0.41, 0.16] | 0.11 | 0.21 [−0.07, 0.50] | 0.18* | 0.36 [0.07, 0.64] | −0.02 | −0.05 [−0.33, 0.24] | −0.05 | −0.11 [−0.39, 0.17] |
| Water bottle + Prompts $n_i$ = 105 $n_c$ = 97 | 0.001 | 0.00 [−0.28, 0.28] | 0.17 * | 0.35 [0.07, 0.63] | 0.03 | 0.06 [−0.22, 0.33] | −0.07 | −0.12 [−0.40, 0.15] | −0.04 | −0.08 [−0.35, 0.20] | 0.11 | 0.22 [−0.06, 0.49] | −0.07 | −0.14 [−0.41, 0.14] | −0.10 | −0.21 [−0.49, 0.07] |
| Water bottle + Goal setting $n_i$ = 113 $n_c$ = 118 | −0.08 | −0.15 [−0.41, 0.10] | 0.19 ** | 0.38 [0.12, 0.64] | −0.003 | −0.01 [−0.27, 0.25] | 0.00 | 0.00 [−0.26, 0.26] | 0.09 | 0.17 [−0.09, 0.43] | 0.10 | 0.20 [−0.06, 0.46] | −0.001 | 0.00 [−0.26, 0.26] | −0.06 | 0.12 [−0.14, 0.38] |
| Water bottle + Feedback $n_i$ = 96 $n_c$ = 107 | −0.16 * | −0.31 [−0.59, −0.04] | 0.004 | 0.01 [−0.27, 0.28] | −0.17 * | −0.34 [−0.61, −0.06] | −0.06 | −0.13 [−0.41, 0.14] | 0.02 | 0.04 [−0.23, 0.32] | 0.10 | 0.19 [−0.08, 0.47] | −0.13 | −0.25 [−0.53, 0.02] | −0.13 * | −0.27 [−0.54, 0.01] |
| Promo video + Prompts $n_i$ = 140 $n_c$ = 93 | 0.09 | 0.17 [−0.09, 0.44] | 0.08 | 0.15 [−0.11, 0.42] | 0.14 ** | 0.28 [0.01, 0.54] | 0.02 | 0.04 [−0.23, 0.30] | −0.03 | −0.07 [−0.33, 0.19] | 0.09 | 0.18 [−0.09, 0.44] | −0.06 | −0.12 [−0.38, 0.15] | −0.13 * | −0.27 [−0.54, −0.01] |
| Promo video + Goal setting $n_i$ = 124 $n_c$ = 89 | 0.002 | 0.00 [−0.27, 0.28] | 0.13 | 0.26 [−0.02, 0.53] | 0.12 | 0.26 [−0.01, 0.53] | 0.08 | 0.16 [−0.12, 0.43] | 0.12 | 0.24 [−0.04, 0.51] | 0.10 | 0.20 [−0.07, 0.47] | 0.01 | 0.02 [−0.25, 0.29] | 0.01 | 0.03 [−0.24, 0.30] |
| Promo video + Feedback $n_i$ = 117 $n_c$ = 89 | −0.05 | −0.02 [−10.81, 10.77] | −0.09 | −0.03 [−10.75, 10.69] | −0.03 | −0.01 [−10.78, 10.76] | 0.01 | 0.01 [−10.78, 10.79] | 0.03 | 0.01 [−10.69, 10.71] | 0.08 | 0.03 [−10.72, 10.78] | −0.13 | −0.06 [−10.87, 10.75] | −0.17 * | −0.06 [−10.76, 10.64] |
| Prompts + Goal setting $n_i$ = 120 $n_c$ = 94 | 0.15 * | 0.30 [0.03, 0.57] | 0.19 ** | 0.38 [0.10, 0.65] | 0.13 * | 0.26 [−0.01, 0.54] | 0.09 | 0.19 [−0.08, 0.46] | −0.05 | −0.10 [−0.37, 0.17] | 0.06 | 0.13 [−0.14, 0.40] | −0.03 | −0.07 [−0.34, 0.20] | −0.03 | −0.06 [−0.32, 0.21] |
| Prompts + Feedback $n_i$ = 115 $n_c$ = 95 | 0.09 | 0.17 [−0.10, 0.44] | −0.01 | −0.02 [−0.30, 0.25] | −0.02 | −0.05 [−0.33, 0.22] | 0.02 | 0.01 [−0.26, 0.29] | −0.11 | −0.23 [−0.50, 0.05] | 0.06 | 0.11 [−0.16, 0.39] | −0.17 * | −0.34 [−0.61, −0.07] | −0.24 *** | −0.49 [−0.77, −0.22] |
| Goal setting + Feedback $n_i$ = 102 $n_c$ = 95 | 0.003 | 0.01 [−0.27, 0.29] | 0.01 | 0.03 [−0.25, 0.31] | −0.06 | −0.12 [−0.40, 0.16] | 0.08 | 0.16 [−0.12, 0.44] | 0.03 | 0.06 [−0.22, 0.34] | 0.04 | 0.08 [−0.20, 0.36] | −0.11 | −0.22 [−0.50, 0.06] | −0.08 | −0.16 [−0.44, 0.12] |

Note. * $p < 0.05$, ** $p < 0.01$, *** $p < 0.001$. $n_i$ = number of participants in the intervention group; $n_c$ = number of participants in the control group. Standardized regression coefficients are provided.

Receiving Prompts in combination with setting up Goals had large effects on Social norm as well as Awareness of need and a moderate effect on Awareness of consequence, an increase in these constructs was observed in the exposure groups, while this was not the case for the control groups. The initial levels of both types of awareness were higher in the control group, compared to the intervention group. Additionally, when combining Goal setting and Prompts, the reverse effect on Behavior becomes insignificant.

The combination of Prompts and Feedback as well as Goal setting and Feedback did not yield any significant positive effects on any of the intervention outcomes. However, when combining Prompts and Feedback as well as Goal setting and Feedback, the negative effects on Awareness of consequence become non-significant. Moreover, when combining Goal setting and Feedback, the reverse effects on Intention and Behavior become insignificant, when it is not the case for the combination of Prompts and Feedback. In fact, the exposure to both Prompts and Feedback resulted in even bigger and large reverse effects on Intentions and Behavior.

In summary, four out of five intervention components had either individual or combined positive effects on at least one of the important program outcomes. Also, we found that reverse effects of providing water bottles on Social norm as well as reverse effects of Prompts on Behavior become insignificant when combined with other intervention components. Moreover, both water bottles and Prompts had clear value for the promotion of the targeted outcomes, especially in combination with other intervention components. From the five intervention components, Feedback was found not to have any positive effects, neither alone nor paired with other components. Furthermore, Feedback yielded undesirable effects that in most cases did not become insignificant with exposure to other intervention components.

## 4. Discussion

The present study employed a factorial experimental design [24,25] in order to test which components of the intervention "Aquatic" aimed at reducing bottled water use are the most effective when presented to a sample of adolescents. Most intervention components show promise to promote factors associated with the reduction of bottled water use. However, not all components of the intervention were found to be equally effective, and some components were found to be effective only when combined with other components. Overall, the many components of the intervention had inconsistent results both individually and in combination with each other, which highlights that one should regard the results of this study with caution and as a guide for future investigation. While some effects show promise, many of the expected outcomes were not observed, highlighting that encouraging adolescents to reduce bottled water consumption is not as straightforward as just enabling their behavior and prompting them to refill their canteens. Further, we will discuss specific relevant findings and their theoretical implications as well as ways forward in improving the "Aquatic" intervention. Significant individual effects will be discussed first, followed by the discussion of combined effects.

Receiving water bottles had a negative effect on social norm, meaning that those individuals who did not receive water bottles tended to assume that using bottled water is more socially acceptable than those individuals, who received water bottles as a part of the intervention. This, however, should not be interpreted as a negative outcome. Receiving a water bottle likely drew the participants' attention to the contrary social norm and thus led to the decrease in the perceived social pressure to use refillable canteens for water. These findings, however, highlight that it may not be effective to implement the intervention in small groups and the intervention has the most potential if implemented universally inside a given group (e.g., the whole school, rather than just a few groups of students). Universal distribution of water bottles would potentially have a positive effect on social norms, since everyone in the reference group would be using canteens for drinking water.

Contrary to what was expected, providing participants with reusable canteens for drinking water did not affect their perceived behavioral control, nor their declared habits.

It may be that participants overestimated their behavioral control prior to the intervention and receiving the means for behavioral change did not affect their perceptions because of a ceiling effect. Additionally, it might be that more time is needed for the intervention to translate into significant changes in reported habits.

The effects of the Promo video were only partially as expected. While the video was primarily intended to target social norms by providing social modeling, this effect was not statistically significant, nor was the additional effect on Awareness of need significant. The video, however, as expected, had a significant positive effect on the participants' awareness of the consequences of their bottled water use. It must be stressed that the video used for the intervention contained a multitude of scenes and therefore its effect might not be as straightforward as one would hope.

Providing Prompts urging to fill the canteen with water had a twofold effect. While those who received the prompts reported higher social norms, indicating that they were more likely to believe that using canteens and refilling them with tap water was socially desirable, the self-reported behavior for this group showed a smaller increase in environmentally friendly behavior, compared to the no-prompts group. It is possible that prompts elicit contrarian behavior [42], which is typical of the investigated age group, and perhaps the frequency and contents of the prompts should be reviewed and further tailored to the age group that receives them. This could be also the reason why we did not observe the expected Prompts effect on the habit of buying bottled water.

Setting a Goal had a significant positive effect on participants' awareness of the need for the reduction of bottled water use. It is likely that by setting a goal, one internalizes it and thus to resolve the resulting cognitive dissonance between a goal and a lack of external proof of its need, one seeks out confirmatory evidence for one's goal. In a sense, by allowing participants to set their goals we make use of their confirmation bias which leads them to seek out other reasons for their behavior [43]. Although, contrary to what was expected, we failed to provide evidence that goal setting has a direct effect on perceived behavioral control, it still could be seen as a promising strategy for targeting phenomena related to behavioral change [7,22].

Providing Feedback, however, had an unexpected significant negative effect on intention and behavior. Moreover, the effect of Feedback on awareness of consequences was contrary to what was expected. Also, Feedback did not have the hypothesized effect on Perceived behavioral control. In particular, those who did not receive feedback showed a bigger increase in intention, behavior, and awareness of consequences, compared to the feedback exposure group. The most likely reason for this is the contents of the feedback, which highlighted the amount of money saved, carbon dioxide reduction, and plastic waste reduction. All of these metrics, while related to bottled water use and societally relevant, are not necessarily individually relevant, these metrics might be too distant or too individually unimpactful [33], which led to them having the opposite effect than intended. The Savannah principle proposes that we are not adapted to think in global terms and things that are too vast or too slow tend not to concern us [44]. In this case, providing feedback might have not been as impressive for adolescents and served more as "noise" rather than making the outcomes of their actions salient, perhaps leading to avoiding behavior. Alternatively, when exposed to this particular feedback, participants may have become more aware of the hard to comprehend and distant outcomes of their behavior, thus discouraging them from acting pro-environmentally. Additionally, providing feedback about money saved might elicit behaviors that are simple monetary decisions, where a person decides that they can afford to waste that amount of money or even engage in costly signaling by using bottled water [45,46]. Those who did not receive feedback likely found personal and more proximal and individually relevant reasons for their behavior, hence the positive increase in their awareness of consequence, intention, and self-reported behavior. In essence, the findings strongly suggest that the Feedback component of the intervention will need to be thoroughly reworked for future applications.

The combined effects of receiving a water bottle and other components of the intervention lend further insight into the functioning of the "Aquatic" intervention. While providing refillable canteens did not have a positive effect on any of the outcome variables, it was revealed that having a canteen resulted in significant additive effects that strengthen the effects of some of the other components of the intervention. This illustrates that the availability of behavior, while not necessarily perceived as creating change, is a catalyst for other factors affecting behavior. Receiving a canteen strengthened the negative effect of Feedback, further showing that having the means of a behavior pushed together with individually irrelevant consequences results in contrary behavior. On the other hand, enabling water refilling behavior directed the effect of the Promo video toward the perceived control of own behavior and the effect of Prompts to positively affect awareness of need, as well as strengthened the positive effect of Goal setting. These findings overall show a synergy between internal (e.g., goals, beliefs) and external (e.g., access to a behavior, ease of behavior) forces affecting pro-environmental behavior, highlighting that effective change requires both.

While on their own Prompts may elicit contrarian behavior, they have a positive effect if coupled with either Promo video, or a Goal setting, suggesting that if one already is invested in pursuing a certain behavior, then reminding one to perform that behavior is effective. This type of reminder likely is not perceived as nagging, but as helping, thus it is not perceived as external pressure, but external help. However, when Prompts are coupled with Feedback, participants likely perceive both these factors as nagging and pressure to behave. The result is that adolescents refuse to follow the suggested behavior as an act of defiance because they have no personal reason for doing so [40].

To sum up, enabling behavior appears to be a crucial part of an intervention targeted at reducing bottled water use. However, along with enabling, one needs to provide tangible and personal reasons for acting in a certain way, creating a synergy between internal and external determinants of behavior. Pressuring individuals into behavior, providing intangible information and personally unimportant feedback tends to have an opposite effect, thus individuals should not be pressured into behavior, but rather provided personally relevant reasons for behavior.

*Strengths, Limitations, and Implications*

Using a factorial experiment for the evaluation of the intervention components is among the obvious strengths of the current study, as it is both a comprehensive and cost-effective strategy for optimizing behavioral interventions [25]. Second, the current study is among the first that aim to explain adolescents' pro-environmental behavior change within the CADM [6] approach. Finally, the study uses the Latent change modelling approach [35] addressing the multilevel structure of the data that allows the precise evaluation of the intervention effects.

Using single-item indicators for the CADM constructs is a limiting factor in this study. In the CADM-related literature, it is common to use multiple indicator measures, as it allows to encompass the multimodality of such constructs as, for example, social norms [6]. On the other hand, it can be argued that, for example, personal norms could be adequately measured with a single "obligation" item [9]. Due to time restrictions, in the current study, we chose single-item indicators for all constructs. However, in future research, we would advocate using latent variables with multiple indicators for measuring the constructs of the CADM, because using single items might to some extent compromise the precision of construct operationalization.

Second, we used self-report measures for evaluating environmentally friendly behavior instead of observing an actual behavior. Although this approach is common for studies within the CADM framework [6], observed measures would strengthen the conclusions regarding the intervention effects on the reduction of bottled water use. Therefore, whenever possible, we encourage trying to find ways to observe actual behavior instead of limiting findings to self-report measures.

Third, as we fully randomized the assignment to four out of five intervention conditions, one of them, particularly, the distribution of reusable water bottles, was performed on a school level by randomly choosing one of two available schools. Randomization at the school level potentially could have affected the baseline level differences in some CADM constructs (e.g., Social norm and Habit) as well as intervention effects, as it is very hard to fully match school-level contexts for the intervention implementation. Our strategy was chosen to avoid spillover effects when the intervention indirectly affects people who have not participated in the program but had contact with people who did [47], however, for future research we would advise full randomization for all intervention components. Additionally, despite the efforts taken to reduce the spillover effects, the factorial experiment was implemented in the community setting (i.e., schools), where the participants not exposed to the particular components still had a possibility to observe the exposure to these components provided to their peers. This could have introduced some bias in the resulting intervention effects.

Fourth, the current study was based on a behavior-specific theoretical model, the CADM. The CADM focuses on specific components that explain the mechanisms of how each specific behavior occurs and allows one to track which factors are most relevant for certain behaviors [6]. Yet besides behavior-specific factors other more general environmental considerations, such as environmental values and self-identity, might play a role too [48–50]. They might give an insight into why certain intervention measures were not effective. For example, adolescents with stronger environmental values might be more susceptible to environmental behavior change interventions. Future intervention studies could expand the current study and consider these factors.

Fifth, we used reusable water canteens made of plastic as one of the intervention components. The design of the canteens was determined by adolescents' preferences in focus groups. Yet the material of the canteens itself could be considered a less environmentally friendly option, even though it seems attractive to participants. Future studies could resolve this limitation by giving reusable canteens only to those participants who do not own a personal one. Thereby reducing the environmental impact of the intervention material. In addition, to ensure uniformity of intervention elements, future studies could provide canteens made of materials with less environmental impact.

Sixth, one could have a positive attitude toward tap water and yet choose bottled water because of certain features (e.g., carbonation or flavors) that are not readily available in tap water. Future studies could control for these variables and test the effect thereof on the effectiveness of intervention elements.

Despite these limitations, the current study provides the evidence basis for the promotion of pro-environmental behavior in adolescence. It implies that there are some means of addressing adolescents' bottled water use that could be relatively easily applied in school, family, or community contexts. However, the current study addresses the optimization stage of the intervention in development. Therefore, as a future step, first, based on the results, we plan to adjust the components of the intervention program "Aquatic" (e.g., Feedback should potentially be removed from the combined set of intervention components). Second, we plan to assess the short-term as well as long-term full intervention effects both on primary and secondary pro-environmental outcomes. Finally, we expect to present the developed intervention to a broader community to further evaluate intervention outcomes in different socio-cultural contexts.

## 5. Conclusions

The present study provided novel evidence on how the CADM can be used in planning and evaluating the intervention program to tackle problematic bottled water use. First, when considering individual effects, no clear pattern emerged on the effectiveness of the tested components. Second, there was a clear pattern of feedback having a negative effect on the program outcomes when in combination with almost all other components. Thus, it is suggested to omit the feedback component from further implementation of the program.

Third, future intervention studies could consider more environmentally friendly options for reusable canteens.

**Supplementary Materials:** The following are available online at https://www.mdpi.com/article/10.3390/su13126758/s1, Table S1: Individual effects' mean intercepts and slopes for intervention and the control groups and the baseline comparison, Table S2: Combined (two by two) effects' mean intercepts and slopes for intervention and the control groups and the baseline comparison.

**Author Contributions:** Conceptualization, I.T.-K.; methodology, I.T.-K. and G.K.; formal analysis, I.T.-K.; investigation, I.T.-K., G.K., M.S.P., A.B., V.G. and L.J.; data curation, I.T.-K. and G.K.; writing—original draft preparation, I.T.-K., G.K. and M.S.P.; writing—review and editing, A.B., V.G., L.J. and M.Ö.; supervision, M.Ö. All authors have read and agreed to the published version of the manuscript.

**Funding:** This research is funded by the European Social Fund according to the activity "Improvement of researchers' qualification by implementing world-class R&D projects". Grant number 09.3.3-LMT-K-712-01-0017.

**Institutional Review Board Statement:** The study was conducted according to the guidelines of the Declaration of Helsinki and approved by the ethical committee of the Department of Psychology of Mykolas Romeris University (decision No. 2/-2020).

**Informed Consent Statement:** Informed consent was obtained from all subjects involved in the study.

**Data Availability Statement:** Data will be made available upon request.

**Acknowledgments:** We would like to thank Oksana Malinauskienė for her contribution in piloting the questionnaire and data collection as well as Rita Žukauskienė for the comments on an earlier draft of this manuscript. In addition, we thank the reviewer for many important comments and suggestions that helped to advance the initial version of the manuscript.

**Conflicts of Interest:** The authors declare no conflict of interest.

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
