# Peer review of "Reducing Bottled Water Use among Adolescents: A Factorial Experimental Approach to Testing the Components of the “Aquatic” Program"

_sustainability, doi:10.3390/su13126758_

Round 1

Reviewer 1 Report

This manuscript addresses a very important issue, reducing bottled water usage among adolescents. The literature review is thorough, the research methods are sound, and the findings are well supported. Additionally, the manuscript is very well written, therefore I have no suggestions for improving the manuscript.

Author Response

#1

This manuscript addresses a very important issue, reducing bottled water usage among adolescents. The literature review is thorough, the research methods are sound, and the findings are well supported. Additionally, the manuscript is very well written, therefore I have no suggestions for improving the manuscript.

  1. Thank you very much for the positive evaluation.

Reviewer 2 Report

The methodology of empirical research should be described in more detail. Especially the methodology of selecting the research sample should be specified. There are no separate conclusions and no final summary of the research results.

Author Response

#2

The methodology of empirical research should be described in more detail. Especially the methodology of selecting the research sample should be specified. There are no separate conclusions and no final summary of the research results.

  1. Thank you for your comment. We have added the concussion section (lines 710-718). The methodology was described in more detail in line with the suggestions of other reviewers.

Reviewer 3 Report

The authors present an interesting study, in which they investigate a complex behaviour-change campaign comprising several measures as a scientific field-experiment. The aim is to quantify effects not only of each measure on its own but also considering their interactions. The behaviour targeted is the consumption of bottled water, which is little investigated but highly relevant from an environmental perspective, due to the waste generated and the consumption of energy and emission of CO2E, particularly for transport. Also, the investigated population, which are adolescents, are under investigated, though, highly relevant. For analysis, a quite sophisticated method (i.e., latent change models) is used. Only few significant and even some negative effects were found.

Overall, the measures were well designed, and a complex sampling plan implemented. Particularly remarkable is the theory-based approach for the intervention design, and the participative tuning of the measures with representatives of the target population. The factorial design of the interventions is impressive; however, the sample appears not to be optimal for such a complex design. Similarly, the sophisticated statistical analyses do not appear to fit to the rather rough measures used to gather the data. The authors run a very large number of tests (120 effects), of which only few (22) are significant on a p < 0.05 level – which is still 3.7 times larger than pure random, but, since no clear patterns emerged, not a solid foundation for general interpretations. Particularly concerning is the fact that, for behaviour, only negative effects turned significant. While the lack of significant effects or negative effects not necessarily reduces the value of the paper, it raises some concerns and makes the discussion appear somewhat overly optimistic.

The main problem of the paper that the authors still can address is the analysis. In my opinion, the approach used by the authors is suboptimal – though, of course, the authors are free to defend their perspective. Starting from the 32 experimental conditions presented in Table 1 (Page 4 to 5), the authors combine groups to identify effects of certain components (single measures of the combinations of two measures). This means, they combine all conditions where, e.g., one measure is present with all groups where the measure is absent. While this leads to larger group sizes for comparison and, thus, more power for the statistical tests, it also leads to tremendous ‘noise’ just by the different other measures applied in each group. The problem with this kind of ‘noise’ is that it not only makes it more difficult to identify effects, it also can lead to biases. Whether an effect is found not only depends on the effect of the measure investigated but also on the effects of the other measures applied. As a result, important effects might not be detected by this procedure; but it is also possible that effects identified are not actually related to the measure investigated. To make things worse, the study design leads to many other sources of noise and potential biases. E.g., effects might not be due to the measures but differences in the sample (e.g., there is no way to distinguish an effect from providing the water bottle from an effect due to differences of the two schools); and the low reliability of the one-item measures leads to additional noise – not to mention the, in general, lower data quality in field studies.

Of course, a larger sample or a slimmer experimental plan would be the perfect solution, but the authors cannot change that now. So, what is needed is to start from the actual experimental groups according to Table 1 – and further investigate influences from sample differences (i.e., effects per experimental group and sample cluster). This, most probably, would lead to very small group sizes, which would not allow for sufficient power to run statistical tests. Nevertheless, it should be possible to identify some similar effects, which also can be theoretically founded and, thus, would allow to combine groups (e.g., if different classes of the schools show similar effects in one experimental condition or if certain measures never show any relevant effect). Hoping that at least some of these groups will be large enough to run at least simple statistical tests, the authors can quantify the effects of specific measures or their combinations. Prototypically, conditions with certain combinations of measures are compared to the group with no measure at all (or no measures that showed any relevant effect in any combination with other effects) – or to the group with all measures to investigate how much the effect is reduced by omitting a combination of measures. However, other comparisons are also possible. For this, I would use as simple methods as possible, considering the small group sizes and low data quality. Such a procedure can be very informative to understand at least the effects of the combinations of measures, for which enough data are available. However, it also can easily be abused to ‘create’ or ‘obscure’ effects as desired. Because of this, all decisions need to be well documented and justified. Of course, this is just a proposal, and the authors might come up with other ideas. The important point is that combining cases with very different ‘experimental manipulations’ might not lead to trustworthy results.

Besides this mayor issue, there are several smaller issues the authors should address by directly improving the mentioned parts or discussing their decisions:

  • The authors base their study on the CADM, which, of course, is a valid decision. Nevertheless, the model, particularly potential shortcomings, might have been discussed. I was particularly surprised that affective aspects (e.g., taste or disgust related to filling a bottle in a public bathroom) were not considered. In our studies on bottled-beverage consumption, these factors were among the most important. And, on a side note, while basing an intervention on a theoretical foundation is great, for most effective interventions, an empirical pre-investigation is required to identify the actually problematic psychological factors.
  • To involve representatives of the target population in the design of the measures is very positive. However, from an environmental perspective, providing a drinking bottle for free, is problematic, since, in many cases, the bottle is not sufficiently often used and, in the end, the campaign might produce more environmental impact than it prevents. Particularly if the study is used as basis for a larger campaign, this problem needs to be mentioned.
  • The behaviour-change techniques (BCTs) were mostly well selected, and the authors invested a lot of effort to implement them in a way that maximizes the effect on the targeted persons and minimizes effects on other participants. However, I do not see the relation of the video as described to social modelling. What is the normative information of this video? As described, it appears only to visualize the (negative) consequences of consuming bottled water.
  • One of the BCT was goal setting. This, often, results to be a very effective measure. However, it involves some challenges for the analysis of effects since this technique can fail in two different forms. Participants might not fulfil the set goal – particularly, when set too high – or they might set such a low goal that no effect results. Of course, both forms are failures and if other techniques lead to more effects, we might not bother about this difference. Nevertheless, it might be interesting to investigate, whether the effectivity of this technique depends (non-linearly) on the goal selected.
  • Table 2 should be designed differently. It appears not so important whether the techniques are applied to 9th or 10th graders, but how many classes in which school received which interventions (i.e., how were the interventions distributed over all classes investigated). Or were there only 4 classes (i.e., one 9th and one 10th grade in each of the two schools – regarding the numbers, this, then, would be very huge classes)?
  • The authors should discuss in more detail possible biases in the estimation of effects due to the sample structure. For example, the effect of distributing bottles cannot be distinguished from a possible effect of differences of the two schools investigated. Further, many ‘spill-over’ or ‘leaking’ effects are still possible, despite the efforts taken by the authors to avoid them. For example, the fact that one measure could change the behaviour, can change the behaviour of other participants not exposed to that BCT, because they observe the behaviour of the other participants. Many of these effects cannot totally be avoided or considered in the analyses. This doesn’t mean that nothing can be learned from these data, but the authors should at least discuss these problems or, even better, address them in the analyses as far as possible.
  • The authors use 1-item measures, which is not too common for psychological investigations and much less for fitting latent-change models. Nevertheless, literature (and my own experience) shows that they work quite well, sometimes even better than multi-item scales. And then again, the items used in this study not always cover the constructs they should represent. E.g., social norms might not only comprise injunctive norms but also descriptive norms and other forms of social pressure. Also measuring habits as “I am used to buying bottled water” might not well assess the construct. PBC is well assessed, but difficulty might be more critical here. Really problematic is the assessment of the behaviour (“I bought bottled water in the last four weeks” - never or almost never (1) to constantly (5)). First, what is meant with ‘constantly’? This can be anything between a bottle every some days to a bottle every minute. Similarly, ‘almost never’ can be five sixpacks in a month. Also, that the item relates to buying (and not consuming) and the very large period of four weeks lowers the information content of data gathered with this item. Assessing the number of bottles (or even litres) in the last couple of days (or on a ‘typical day’) would have been a better option. While the authors cannot change this, anymore, it must be considered in the interpretation / discussion of the results that the behaviour is assessed very imprecisely.
  • Related to the problem just mentioned is that the authors sometimes use different concepts for the interpretation of the results than used for operationalizing the constructs. Take, e.g., Lines 538 to 540: “Receiving water bottles had a negative effect on social norm, meaning that those individuals who did not receive water bottles tended to assume that using bottled water is more prevalent than those individuals, who received water bottles as a part of the intervention.” Norm was assessed as injunctive norm (i.e., whether important others expect the participants not to buy bottled water), not as descriptive norm. Thus, the data does not provide any information about perceived prevalence of the behaviour (which, indeed, should have been assessed, too). This reference to prevalence is found also in other parts (e.g., Line 565).
  • In the analyses, the authors mix-up testing of a-priory hypotheses and exploratory search for possible effects. It should be made clearer, what effects are used for testing the hypotheses – and whether the data supports them. For the exploratory analyses, due to the large number of tests, the authors should consider correcting the significance level for the number of tests. Further, the authors should check (and report) whether floor/ceiling effects might affect the estimation of the effects.
  • The interpretation and discussion of the results needs to be much improved – though, as explained above, I would recommend running different analyses requiring completely new interpretations and discussions. The problem with the current interpretation is that the authors just pick out some single significant results and derive quite general conclusions from them. However, from such a complex and comprehensive experimental design, any relevant effect should lead to a pattern in the results. If a measure shows only in one specific combination an effect but not in the other about 12 combinations, I would be very cautious in interpreting this as a relevant effect. Further, the discussion is much too optimistic regarding the results. The results clearly indicate a failure of the campaign (e.g., all effects on behaviour were negative). This is not necessarily a bad thing for a scientific investigation and even for praxis such failure can be a valuable starting point to improve interventions. But for this, the failure must be accepted, and it must be reflected, what went wrong. The feedback intervention is a nice example for this (Lines 581 to 602): This measure systematically failed, which the authors try to explain with rather adventurous processes. The straightforward explanation that the impact is simply too small to impress the participants is avoided.
  • And, finally, some issues with the wording: Often, the authors specify something like “tap water (or alternatively well or spring instead of purchasing bottled water)”. This makes the text in some parts long and more difficult to read. It would be better to specify this once and then only to talk about tap water (or not-bottled water). Further, the term ‘pre-test’ (e.g., Line 318 or 322) might be misleading as a pre-test could also be understood as a pilot before the actual study. I would suggest ‘pre-intervention’ and ‘post-intervention’ measurement.

Author Response

#3

The authors present an interesting study, in which they investigate a complex behaviour-change campaign comprising several measures as a scientific field-experiment. The aim is to quantify effects not only of each measure on its own but also considering their interactions. The behaviour targeted is the consumption of bottled water, which is little investigated but highly relevant from an environmental perspective, due to the waste generated and the consumption of energy and emission of CO2E, particularly for transport. Also, the investigated population, which are adolescents, are under investigated, though, highly relevant. For analysis, a quite sophisticated method (i.e., latent change models) is used. Only few significant and even some negative effects were found.

Overall, the measures were well designed, and a complex sampling plan implemented. Particularly remarkable is the theory-based approach for the intervention design, and the participative tuning of the measures with representatives of the target population. The factorial design of the interventions is impressive; however, the sample appears not to be optimal for such a complex design. Similarly, the sophisticated statistical analyses do not appear to fit to the rather rough measures used to gather the data. The authors run a very large number of tests (120 effects), of which only few (22) are significant on a p < 0.05 level – which is still 3.7 times larger than pure random, but, since no clear patterns emerged, not a solid foundation for general interpretations. Particularly concerning is the fact that, for behaviour, only negative effects turned significant. While the lack of significant effects or negative effects not necessarily reduces the value of the paper, it raises some concerns and makes the discussion appear somewhat overly optimistic.

The main problem of the paper that the authors still can address is the analysis. In my opinion, the approach used by the authors is suboptimal – though, of course, the authors are free to defend their perspective. Starting from the 32 experimental conditions presented in Table 1 (Page 4 to 5), the authors combine groups to identify effects of certain components (single measures of the combinations of two measures). This means, they combine all conditions where, e.g., one measure is present with all groups where the measure is absent. While this leads to larger group sizes for comparison and, thus, more power for the statistical tests, it also leads to tremendous ‘noise’ just by the different other measures applied in each group. The problem with this kind of ‘noise’ is that it not only makes it more difficult to identify effects, it also can lead to biases. Whether an effect is found not only depends on the effect of the measure investigated but also on the effects of the other measures applied. As a result, important effects might not be detected by this procedure; but it is also possible that effects identified are not actually related to the measure investigated. To make things worse, the study design leads to many other sources of noise and potential biases. E.g., effects might not be due to the measures but differences in the sample (e.g., there is no way to distinguish an effect from providing the water bottle from an effect due to differences of the two schools); and the low reliability of the one-item measures leads to additional noise – not to mention the, in general, lower data quality in field studies.

Of course, a larger sample or a slimmer experimental plan would be the perfect solution, but the authors cannot change that now. So, what is needed is to start from the actual experimental groups according to Table 1 – and further investigate influences from sample differences (i.e., effects per experimental group and sample cluster). This, most probably, would lead to very small group sizes, which would not allow for sufficient power to run statistical tests. Nevertheless, it should be possible to identify some similar effects, which also can be theoretically founded and, thus, would allow to combine groups (e.g., if different classes of the schools show similar effects in one experimental condition or if certain measures never show any relevant effect). Hoping that at least some of these groups will be large enough to run at least simple statistical tests, the authors can quantify the effects of specific measures or their combinations. Prototypically, conditions with certain combinations of measures are compared to the group with no measure at all (or no measures that showed any relevant effect in any combination with other effects) – or to the group with all measures to investigate how much the effect is reduced by omitting a combination of measures. However, other comparisons are also possible. For this, I would use as simple methods as possible, considering the small group sizes and low data quality. Such a procedure can be very informative to understand at least the effects of the combinations of measures, for which enough data are available. However, it also can easily be abused to ‘create’ or ‘obscure’ effects as desired. Because of this, all decisions need to be well documented and justified. Of course, this is just a proposal, and the authors might come up with other ideas. The important point is that combining cases with very different ‘experimental manipulations’ might not lead to trustworthy results.

R: Thank you very much for the positive evaluation and the thoughtful insights regarding the data analytic strategy. Nevertheless, we do believe that the strategy we chose is well justified in the methodological literature. As we stated in lines 159-161 we have used a factorial experiment for the intervention optimization stage, namely, the evaluation of individual and combined intervention components, as it is a comprehensive method that is most appropriate when conducting research to select components for inclusion in a multicomponent intervention and it allows doing it with sufficient statistical power with a reasonable sample size (Collins, Dziak, & Li, 2009, Collins, Dziak, Kugler, & Trail, 2014). This means that the study design was developed by already planning to analyze data as it was suggested to be most appropriate for the factorial experiments. However, as by reading your comment, we realized that this methodological approach, although presented in the introduction, may not have been sufficiently addressed in the methods section. Therefore, we have now provided additional information regarding the methodological approach leading to the data analysis strategy chosen for the presentation of the study results (see lines 159-161, 171-176 as well as provided additional clarification note for the Table 1).

Besides this mayor issue, there are several smaller issues the authors should address by directly improving the mentioned parts or discussing their decisions:

  • The authors base their study on the CADM, which, of course, is a valid decision. Nevertheless, the model, particularly potential shortcomings, might have been discussed. I was particularly surprised that affective aspects (e.g., taste or disgust related to filling a bottle in a public bathroom) were not considered. In our studies on bottled-beverage consumption, these factors were among the most important. And, on a side note, while basing an intervention on a theoretical foundation is great, for most effective interventions, an empirical pre-investigation is required to identify the actually problematic psychological factors.

R. Thank you for this good suggestion. We would frame this idea in a little bit different manner. We wouldn't consider the CADM model as having shortcomings, but rather say that the model is specifically designed to address behavior specific components and uncover behavior specific mechanisms that lead to environmental behavior (Balunde, Jovarauskaite & Poškus. 2020). Yet, this behavior specific focus does not consider other potential factors such as one’s environmental values and environmental self-identity, for example. It has been established that these constructs are important in explaining environmental behavior indirectly via intermediate variables such as moral obligations to perform certain environmental behaviors. We now acknowledge this as a potential limitation (lines 682-687). We agree that negative feelings towards tap water is very important, thank you for pointing this out that we missed to include this information. Indeed we have included this control question in our preliminary questionnaire and we have controlled for those feelings. In our sample 97.5% participants responded with neutral or positive feelings. We have included this information in the method section lines 193-195

  • To involve representatives of the target population in the design of the measures is very positive. However, from an environmental perspective, providing a drinking bottle for free, is problematic, since, in many cases, the bottle is not sufficiently often used and, in the end, the campaign might produce more environmental impact than it prevents. Particularly if the study is used as basis for a larger campaign, this problem needs to be mentioned.

R. Thank you for pointing out this valid aspect. We certainly agree that the environmental impact of intervention targeted at environmental behavior should be minimized as much as possible. For example, by giving reusable canteens only for those participants who doesn’t own one. Yet we also argue that in intervention studies it is important to ensure that all intervention elements that are presented to participants are as uniform as possible. This principle of uniformity could be violated if we would choose a more environmentally-friendly intervention approach towards reusable canteens. Specifically, one won’t be able to distinguish the reasons why this certain element of the intervention was effective for some groups yet not for others (group who received canteens vs group that own their personal canteens).

Before deciding which bottle design will be used for the intervention, we asked for adolescents’ opinions in focus groups. Some more environmentally friendly options (i.e. metal/glass canteens) were not considered as attractive. Final decision was based on adolescents’ preferences. Our concern was that if the target group will see this particular element of intervention as unattractive, they won’t engage in activities related to it and it will compromise the aims of the intervention. Even considering the above we agree that future studies should seek for more environmentally friendly intervention components. We acknowledge this limitation in the manuscript (lines 692-698) and provide our suggestions for future studies.

  • The behaviour-change techniques (BCTs) were mostly well selected, and the authors invested a lot of effort to implement them in a way that maximizes the effect on the targeted persons and minimizes effects on other participants. However, I do not see the relation of the video as described to social modelling. What is the normative information of this video? As described, it appears only to visualize the (negative) consequences of consuming bottled water.

R: We are grateful for your valuable insights, however, like we state in the manuscript, the video was “targeted to increase the awareness of the harm of single-use plastic and modelling possible alternatives to it. The particular video was chosen as it has a humorous component as well as the message is brought through the imitation of romantic relationships which makes it more attractive to adolescents.”. The description, in our opinion, clearly underscores the positive and social modeling side as it provides alternative behaviors (reusable canteen use and all the benefits that comes with it) more, then it visualizes the negative consequences. We have added the additional information on the reusable canteens modeling in the video.

  • One of the BCT was goal setting. This, often, results to be a very effective measure. However, it involves some challenges for the analysis of effects since this technique can fail in two different forms. Participants might not fulfil the set goal – particularly, when set too high – or they might set such a low goal that no effect results. Of course, both forms are failures and if other techniques lead to more effects, we might not bother about this difference. Nevertheless, it might be interesting to investigate, whether the effectivity of this technique depends (non-linearly) on the goal selected.

R: Thank you for this valuable insight. Indeed, it would be interesting to go deeper in reasons why each of the components was effective and why not. Although we do agree that this question is very interesting, we believe it is out of the scope of the current study.

  • Table 2 should be designed differently. It appears not so important whether the techniques are applied to 9th or 10th graders, but how many classes in which school received which interventions (i.e., how were the interventions distributed over all classes investigated). Or were there only 4 classes (i.e., one 9th and one 10th grade in each of the two schools – regarding the numbers, this, then, would be very huge classes)

    R: The information on 9th and 10th graders are provided because (as we stated in the 284-286 lines) the identical randomization procedures were applied to these two cohorts separately. The randomization process and the condition distribution were performed in three levels (lines 265 - 290) and it is more complicated than simply distributing it on the classes level. We balanced the classes number to be equal across all conditions. For the feedback and goal setting conditions the randomization was implemented at the classroom level, thus there is no sense providing the number of classrooms, while only half or quarter of each class received the condition. The average size of the class was 25 adolescents, thus approx. 10 in each condition all the time (except for feedback and goal setting as half of the class in each class received it). In line with the conventional intervention studies reporting standard, we reported the number of participants (not classes) in each condition.

  • The authors should discuss in more detail possible biases in the estimation of effects due to the sample structure. For example, the effect of distributing bottles cannot be distinguished from a possible effect of differences of the two schools investigated. Further, many ‘spill-over’ or ‘leaking’ effects are still possible, despite the efforts taken by the authors to avoid them. For example, the fact that one measure could change the behaviour, can change the behaviour of other participants not exposed to that BCT, because they observe the behaviour of the other participants. Many of these effects cannot totally be avoided or considered in the analyses. This doesn’t mean that nothing can be learned from these data, but the authors should at least discuss these problems or, even better, address them in the analyses as far as possible.

R: Thank you, to address this comment, we have advanced the limitation section. As we have addressed the non-randomization of the “water bottle” component in the previous version of the manuscript, we have now elaborated on possible spillover of other components as well. The limitation addressing these issues now reads as follows: “Third, as we fully randomized the assignment to four out of five intervention conditions, one of them, particularly, the distribution of reusable water bottles, was performed on a school level by randomly choosing one of two available schools. Randomized at school level potentially could have affected the baseline level differences in some CADM constructs (e.g., Social norm and Habit) as well as intervention effects, as it is very hard to fully match school-level contexts for the intervention implementation. Our strategy was chosen to avoid spillover effects when the intervention indirectly affects people who have not participated in the program but had a contact with people who did (Cook and Campbell, 1979), however, for future research we would advise full randomization for all intervention components. Additionally, despite the efforts taken to reduce the spillover effects, the factorial experiment was implemented in the community setting (i.e., schools), where the participants not exposed to the particular components still had a possibility to observe the exposure to these components provided to their peers. This could have introduced some bias in resulting intervention effects.”

  • The authors use 1-item measures, which is not too common for psychological investigations and much less for fitting latent-change models. Nevertheless, literature (and my own experience) shows that they work quite well, sometimes even better than multi-item scales. And then again, the items used in this study not always cover the constructs they should represent. E.g., social norms might not only comprise injunctive norms but also descriptive norms and other forms of social pressure. Also measuring habits as “I am used to buying bottled water” might not well assess the construct. PBC is well assessed, but difficulty might be more critical here. Really problematic is the assessment of the behaviour (“I bought bottled water in the last four weeks” - never or almost never (1) to constantly (5)). First, what is meant with ‘constantly’? This can be anything between a bottle every some days to a bottle every minute. Similarly, ‘almost never’ can be five sixpacks in a month. Also, that the item relates to buying (and not consuming) and the very large period of four weeks lowers the information content of data gathered with this item. Assessing the number of bottles (or even litres) in the last couple of days (or on a ‘typical day’) would have been a better option. While the authors cannot change this, anymore, it must be considered in the interpretation / discussion of the results that the behaviour is assessed very imprecisely.

R: We agree with the reviewer’s critique regarding single use items. Yet, the decision was indeed mindful given that participants of the current study were adolescents. The motivation behind this choice is that we seek to reduce the risk of participants' fatigue and feeling of boredom as much as possible. In addition, we had strict time constraints set by schools. We acknowledge this in the Limitations section (lines 659-661).

  • Related to the problem just mentioned is that the authors sometimes use different concepts for the interpretation of the results than used for operationalizing the constructs. Take, e.g., Lines 538 to 540: “Receiving water bottles had a negative effect on social norm, meaning that those individuals who did not receive water bottles tended to assume that using bottled water is more prevalent than those individuals, who received water bottles as a part of the intervention.” Norm was assessed as injunctive norm (i.e., whether important others expect the participants not to buy bottled water), not as descriptive norm. Thus, the data does not provide any information about perceived prevalence of the behaviour (which, indeed, should have been assessed, too). This reference to prevalence is found also in other parts (e.g., Line 565).

R: We corrected the discussion to refer to socially desirable behavior as opposed to a descriptive norm (lines 551, 554, 575).

  • In the analyses, the authors mix-up testing of a-priory hypotheses and exploratory search for possible effects. It should be made clearer, what effects are used for testing the hypotheses – and whether the data supports them. For the exploratory analyses, due to the large number of tests, the authors should consider correcting the significance level for the number of tests. Further, the authors should check (and report) whether floor/ceiling effects might affect the estimation of the effects.

R: Thank you for the valuable insights. To increase the readability, we present the intervention effects “by components” instead of “by hypothesis”. However, we do state whether the hypotheses were confirmed in the results section, and the exploratory analysis is provided as additional. We appreciate the note regarding the floor/ceiling effect. In the revised version of the manuscript, we have now provided the information regarding the skewness and kurtosis, which could be indicative of possible floor/ceiling effects. It reads as follows: “All variables across two measurement points were approximately normally distributed, as the coefficients of skewness and kurtosis were within the range of ±2 (Gravetter & Wallnau, 2014).”

  • The interpretation and discussion of the results needs to be much improved – though, as explained above, I would recommend running different analyses requiring completely new interpretations and discussions. The problem with the current interpretation is that the authors just pick out some single significant results and derive quite general conclusions from them. However, from such a complex and comprehensive experimental design, any relevant effect should lead to a pattern in the results. If a measure shows only in one specific combination an effect but not in the other about 12 combinations, I would be very cautious in interpreting this as a relevant effect. Further, the discussion is much too optimistic regarding the results. The results clearly indicate a failure of the campaign (e.g., all effects on behaviour were negative). This is not necessarily a bad thing for a scientific investigation and even for praxis such failure can be a valuable starting point to improve interventions. But for this, the failure must be accepted, and it must be reflected, what went wrong. The feedback intervention is a nice example for this (Lines 581 to 602): This measure systematically failed, which the authors try to explain with rather adventurous processes. The straightforward explanation that the impact is simply too small to impress the participants is avoided.

 R: Thank you for this comment. We do not want to deviate from our initially planned analysis procedures, yet we now explicitly state that our intervention produced largely insignificant results. We also amended our interpretations of the effects of feedback with a simpler explanation.

  • And, finally, some issues with the wording: Often, the authors specify something like “tap water (or alternatively well or spring instead of purchasing bottled water)”. This makes the text in some parts long and more difficult to read. It would be better to specify this once and then only to talk about tap water (or not-bottled water). Further, the term ‘pre-test’ (e.g., Line 318 or 322) might be misleading as a pre-test could also be understood as a pilot before the actual study. I would suggest ‘pre-intervention’ and ‘post-intervention’ measurement.

R: Thank you for the suggestions, we have shortened the paragraph by removing the repetitive sentence and have revised the manuscript for similar issues. The methodological terms “pre-test” and “post-test” are widely used in the methodological literature (e.g. Alessandri et al., 2017), when assessing the intervention effects. Therefore, we hope you will understand our choice to stick to these terms

Reference

Alessandri, G., Zuffianò, A., & Perinelli, E. (2017). Evaluating Intervention Programs with a Pretest-Posttest Design: A Structural Equation Modeling Approach. Frontiers in Psychology, 8, 223. https://doi.org/10.3389/fpsyg.2017.00223

Balundė, A., Jovarauskaitė, L., & Poškus, M. S. (2020). Exploring adolescents’ waste prevention via Value-Identity-Personal norm and Comprehensive Action Determination Models. Journal of Environmental Psychology, 72, 101526. https://doi.org/10.1016/j.jenvp.2020.101526

Round 2

Reviewer 3 Report

The authors included some minor changes to the manuscript that lead to improvements. Unfortunately, they did not improve the analyses. My suggestions were mainly meant to increase effect sizes by reducing the within-group variance, in hope for identifying some generalizable patterns in the results beyond randomly significant effects due to the large number of tests. However, the approach used by the authors is not wrong, it is, in my opinion, just suboptimal regarding the small sample and complex experimental design. So, if the authors want to stay with this approach, it is o.k. Then, however, they must accept that close to no significant effects could be found, and the interventions failed regarding changing the target behaviour. To provide at least a small glimpse on what is happening, I would suggest providing the mean changes of all variables for each of the intervention combinations compiled in Table 1 with their estimation errors.

Regarding the minor issues, here my comments to the revisions and replies of the authors:

  • Regarding my point on affective factors, the authors added some important information. However, the main issue is the question, whether the participants like more drinking bottled water, e.g., because they prefer sparkling water, or whether they prefer more drinking tap-water (or other forms of non-bottled water). Even a person that has, e.g., neutral feelings for tap water, might have more positive feelings for bottled water and, thus, prefer the latter. This issue is, of course, no fundamental error of the study, as any study misses out on some aspects, but it might be good to mention this in the discussion so that future researchers consider this.
  • The issue with distributing canteens is well addressed, now. Note however, that it is not so much the material that is an issue but distributing something that is almost never used. In fact, the plastic canteen might be less environmental damaging than a glass or aluminium canteen, if it is thrown away right after production, as the production of glass and aluminium needs much more energy, particularly if no recycled material is used. But this just as a sidenote.
  • That goal setting is not further investigated is o.k.
  • The reply to the issues regarding sample description is not satisfying. The study builds on a nested sample and, for the interpretation of the results, it is essential to know how the sample was structured – and how the effects varied within and between clusters. Of course, this description not necessarily needs to be according to my suggestions; but somehow the authors need to inform the readers about the sample structure. At the very least, the total number of classes in each school and level needs to be provided. If the design could be perfectly balanced regarding all conditions, this information should be provided and would allow to describe the sample structure in a simple form.
  • The text regarding spillover and leaking effects is o.k., even though the authors could have mentioned additionally that these processes usually reduce effect sizes. Thus, such effects could be an explanation for the weak effects found.
  • The interpretation of the variables appears, now, to be according to their operationalization. However, most of the presented effects are, most probably, just randomly significant results due to the high number of tests conducted, i.e., will not be significant anymore once the required correction for multiple testing is applied (see next point).
  • The main problem with the significance testing was not addressed. The authors can mix a-priori hypothesis testing and explorative testing. Then, however, all tests need to be considered explorative regarding the correction of the significance level. The latter is the main issue the authors need to address but ignored in the revision. Thus, it is o.k. to continue with the giant exploration, but the significance level needs to be corrected due to the large number of tests (e.g., using the Bonferroni method, with 120 tests and a significance level of p < 0.05, the values for p of each single test need to be p < 0.05 / 120 = 0.00042 – or, likewise, the confidence intervals need to represent the 0.042% CI instead of the 5% CI). Derived from the current results, only one effect might be actually significant (the reduction of the target behaviour by prompts and feedback). Thus, also the results, discussion and abstract need to be adapted to the fact, that close to no effects are significant.
  • The discussion was considerably improved. However, due to most of the discussed effects, most probably, will not be significant anymore, once the required correction for multiple testis is applied, much of the discussion of the found effects can be omitted.
  • The wording is now o.k.

To resume, the manuscript is o.k., even though, with the method used and the correction for multiple testing applied, most probably, there will be close to no significant effects. Thus, not much can be learned from this study. If this is o.k. for the editors, only the following minor revisions are required:

  • Mention to consider affective aspects such as taste or preference for sparkling water in future studies.
  • The sample structure needs to be explained. At the very least, the total number of classes needs to be reported.
  • Correct the p threshold for multiple testing and only interpret results with p below this corrected threshold as non-random effects. Adapt the results, discussion and abstract accordingly.

Author Response

The authors included some minor changes to the manuscript that lead to improvements. Unfortunately, they did not improve the analyses. My suggestions were mainly meant to increase effect sizes by reducing the within-group variance, in hope for identifying some generalizable patterns in the results beyond randomly significant effects due to the large number of tests. However, the approach used by the authors is not wrong, it is, in my opinion, just suboptimal regarding the small sample and complex experimental design. So, if the authors want to stay with this approach, it is o.k. Then, however, they must accept that close to no significant effects could be found, and the interventions failed regarding changing the target behaviour. To provide at least a small glimpse on what is happening, I would suggest providing the mean changes of all variables for each of the intervention combinations compiled in Table 1 with their estimation errors.

R: Thank you for the suggestion, however, having in mind the small sample sizes in 32 subgroups and the methodological approach we use in the current study, we hope you will understand our choice not to expand the results we are reporting the current manuscript.

Regarding the minor issues, here my comments to the revisions and replies of the authors:

  • Regarding my point on affective factors, the authors added some important information. However, the main issue is the question, whether the participants like more drinking bottled water, e.g., because they prefer sparkling water, or whether they prefer more drinking tap-water (or other forms of non-bottled water). Even a person that has, e.g., neutral feelings for tap water, might have more positive feelings for bottled water and, thus, prefer the latter. This issue is, of course, no fundamental error of the study, as any study misses out on some aspects, but it might be good to mention this in the discussion so that future researchers consider this.

R: Thank you for pointing this out. We have added this recommendation for future studies (lines 710-713).

  • The issue with distributing canteens is well addressed, now. Note however, that it is not so much the material that is an issue but distributing something that is almost never used. In fact, the plastic canteen might be less environmental damaging than a glass or aluminium canteen, if it is thrown away right after production, as the production of glass and aluminium needs much more energy, particularly if no recycled material is used. But this just as a sidenote.
  • That goal setting is not further investigated is o.k.
  • The reply to the issues regarding sample description is not satisfying. The study builds on a nested sample and, for the interpretation of the results, it is essential to know how the sample was structured – and how the effects varied within and between clusters. Of course, this description not necessarily needs to be according to my suggestions; but somehow the authors need to inform the readers about the sample structure. At the very least, the total number of classes in each school and level needs to be provided. If the design could be perfectly balanced regarding all conditions, this information should be provided and would allow to describe the sample structure in a simple form.

R: We have addressed this issue and provided required information (lines 273-275 and 290-291).

  • The text regarding spillover and leaking effects is o.k., even though the authors could have mentioned additionally that these processes usually reduce effect sizes. Thus, such effects could be an explanation for the weak effects found.
  • The interpretation of the variables appears, now, to be according to their operationalization. However, most of the presented effects are, most probably, just randomly significant results due to the high number of tests conducted, i.e., will not be significant anymore once the required correction for multiple testing is applied (see next point).
  • The main problem with the significance testing was not addressed. The authors can mix a-priori hypothesis testing and explorative testing. Then, however, all tests need to be considered explorative regarding the correction of the significance level. The latter is the main issue the authors need to address but ignored in the revision. Thus, it is o.k. to continue with the giant exploration, but the significance level needs to be corrected due to the large number of tests (e.g., using the Bonferroni method, with 120 tests and a significance level of p < 0.05, the values for p of each single test need to be p < 0.05 / 120 = 0.00042 – or, likewise, the confidence intervals need to represent the 0.042% CI instead of the 5% CI). Derived from the current results, only one effect might be actually significant (the reduction of the target behaviour by prompts and feedback). Thus, also the results, discussion and abstract need to be adapted to the fact, that close to no effects are significant.

R: We appreciate such a thorough approach very much, however, with all due respect, we cannot agree with this suggestion. The reviewer would be absolutely right if we were to be looking for random effects in the data pool. However, this is not the case with the study presented in the current manuscript. We did formulate specific hypotheses, that were based on previous findings. However, the field is still growing and, as also mentioned when grounding the study in previous research, a lot still needs to be done to investigate the CADM model in adolescent samples, therefore, not all of the links were to be grounded in advance. However, the CADM constructs are highly interrelated and, therefore, the same means (or intervention components) may affect several of them at once. Testing the effects of all components on all CADM constructs we see as a strength, not as a limitation of the study. Therefore, we do believe that choosing the significance level of p < .05 or CI of 95% is no methodological error and we hope you will understand our choice to stick to the current data analytic and interpretation strategy. On the other hand, we do agree that counting exclusively on p values is not without biases. Therefore, we edited the manuscript to reflect our confidence more on effect sizes and their CI’s (Nakagawa, 2004) which are supporting our current interpretation (see results section). Moreover, to further reflect on the suggestion to correct the p value, the Bonferroni correction is known to be even more biased especially with small effect sizes in this way rejecting truly meaningful, even though small, effects.

Reference

Nakagawa, S. (2004). A farewell to Bonferroni: the problems of low statistical power and publication bias. Behavioral Ecology, 6(15), 1044-1045, https://doi.org/10.1093/beheco/arh107

  • The discussion was considerably improved. However, due to most of the discussed effects, most probably, will not be significant anymore, once the required correction for multiple testis is applied, much of the discussion of the found effects can be omitted.
  • The wording is now o.k.

To resume, the manuscript is o.k., even though, with the method used and the correction for multiple testing applied, most probably, there will be close to no significant effects. Thus, not much can be learned from this study. If this is o.k. for the editors, only the following minor revisions are required:

  • Mention to consider affective aspects such as taste or preference for sparkling water in future studies.
  • The sample structure needs to be explained. At the very least, the total number of classes needs to be reported.
  • Correct the p threshold for multiple testing and only interpret results with p below this corrected threshold as non-random effects. Adapt the results, discussion and abstract accordingly.

R: We have addressed these issues in previous comments